# Experimental Induction of Intracranial Aneurysms in Rats: A New Model Utilizing a Genetic Modification within the EDNRA Gene

**DOI:** 10.3390/brainsci12091239

**Published:** 2022-09-14

**Authors:** Tim Lampmann, Valeri Borger, Jürgen Konczalla, Suzana Gispert, Georg Auburger, Hartmut Vatter, Erdem Güresir

**Affiliations:** 1Department of Neurosurgery, University Hospital Bonn, 53127 Bonn, Germany; 2Department of Neurosurgery, University Hospital Frankfurt, 60528 Frankfurt am Main, Germany; 3Experimental Neurology, Medical Faculty, Goethe University, 60590 Frankfurt am Main, Germany

**Keywords:** intracranial aneurysm, risk factors, EDNRA, endothelin, animal model, rat, basilar artery

## Abstract

The rupture of an intracranial aneurysm (IA) leads to life-threatening subarachnoid hemorrhage. Aside from well-established risk factors, recently published genome-wide association studies of IA revealed the strong association of a common variant near the endothelin receptor type A (EDNRA) gene with IA risk. However, the role of EDNRA in the pathogenesis of IA remains unclear. The aim of this study was to investigate the influence of a genetic modification within the EDNRA gene on IA pathogenesis in a novel in vivo model. Adult wild-type Sprague–Dawley rats (WT rats) and genetically modified rats (EDNRA rats) were used for the induction of IA using arterial hypertension (HT). Animals were stratified into four groups: WT rats without (WT_CTL) and with induction of HT (WT + HT), as well as EDNRA rats without (EDNRA_CTL) and with induction of HT (EDNRA + HT). Blood pressure (BP) was observed for 12 weeks. After the observation period, cerebral arteries were analyzed for morphological (i.e., aneurysmal) changes as well as histological and functional changes by immunofluorescence and functional investigation. In the groups of rats with induction of HT, BP was higher in EDNRA + HT compared with that in WT + HT. No IAs were observed in WT_CTL and EDNRA_CTL but were found in WT + HT and EDNRA + HT. There was no histological difference in the immunofluorescence of EDNRA between all groups. Contractility and potency of endothelin-1 differed between the groups in functional investigation. In summary, we created a new model that is suitable for further studies for better understanding of the role of EDNRA in IA pathogenesis.

## 1. Introduction

Subarachnoid hemorrhage (SAH)—due to a ruptured intracranial aneurysm (IA)—is a fatal disease with high morbidity and mortality [1]. Although the case fatality decreased in the past decades due to better diagnosis and therapeutic options, the impact especially on the middle-aged population is still making it a socioeconomic problem due to the loss of many years of productive life [1,2,3]. There is an overall prevalence of IA by approximately 3.2% [4]. Risk factors such as genetic predisposition, arterial hypertension (HT), and nicotine abuse were already identified for IA development, growth, and rupture [5,6,7,8,9,10]. There is growing evidence that morphological and molecular changes by endothelial cell dysfunction and inflammatory reaction due to hemodynamic stress, caused by wall shear stress (WSS) and HT, are the most important factors of IA formation [11,12,13,14]. For the investigation of IA pathogenesis, in vivo models were established and confirmed that an increased hemodynamic stress such as WSS and HT leads to an increased IA formation [11,15,16,17].

Furthermore, single-nucleotide polymorphisms near the endothelin receptor type A (EDNRA) gene were associated with an increased IA risk [18,19,20]. Endothelin-1 (ET-1) is the most potent vasoconstrictor through its activation of EDNRA, but the endothelin system is complex due to its different reactions depending on the addressed receptor type and multiple ligands [21,22]. The influence of ET-1 and EDNRA on HT and remodeling of vascular wall has already been demonstrated [23,24]. Drug therapy already yielded promising results particularly in animal trials, even though clinical benefit could not be achieved [25,26,27]. Therefore, overall, the role and mechanisms of ET-1 and EDNRA depend on interactions with other underlying factors of IA pathogenesis and remain ambiguous.

Most of the used in vivo models utilize a combination of surgical preparation and additional drug intake (e.g., NaCl, β-aminopropionitrile (BAPN), elastase, corticosterone, or angiotensin) [17]. Especially BAPN and elastase weaken the vessel wall, not allowing further histological investigations of a natural IA development. We hypothesize that the combination of HT and modification within the EDNRA gene aggravates the HT in such a way that IA development is induced—making the intake of interfering drugs superfluous. This may offer an opportunity for further investigations in a natural setting without the alteration of the vessel wall.

Therefore, the aim of this study was to establish a new in vivo model on IA pathogenesis in hypertensive wild-type and within the EDNRA gene-modified rats. We first performed investigations to explore the influence of this genetic modification on IA pathogenesis.

## 2. Materials and Methods

### 2.1. Animals

Adult wild-type Sprague–Dawley (SD) rats (WT rats) and genetically modified SD rats with a heterozygous deletion of 412 base pairs (bp) at 3922–4333 bp within the EDNRA gene (EDNRA rats)—where only the first 4 amino acids MGVL of the EDNRA protein can be synthesized, followed by frameshift and aborted translation—were used for the following experiments. Animals had been generated with ComproZr™ Zinc finger nucleases (SAGE Labs, St. Louis, MO, USA) targeting the DNA sequence GTCATTGATCTCCCCATCAATGTGTTTAAGGTAGGA within the EDRNA genomic sequence (NCBI Ref. Seq.: NC_005118.2). This approach produced 4 founder animals (475 bp deletion at 4315–4789 bp; 7 bp deletion with 1 bp insertion (C) at 4317–4323 bp; 17 bp deletion at 4321–4337 bp; and 412 bp deletion at 3922–4333 bp). The latter deletion covered most of the EDNRA reading frame, so this mutant line was chosen for extensive characterization. The animals were kept under the name SD-*Ednra^em2Sage^* (RRID:RGD_152995518) in the “Haus für Experimentelle Therapie” of the University Hospital Bonn. Animal experiments followed the ARRIVE guidelines [28] and were approved by the North Rhine Westphalia State Agency for Nature, Environment and Consumer Protection (AZ 84-02.04.2012.A449). All local and national guidelines for care and use of laboratory animals were followed.

Animals were randomized into four groups: WT rats without (WT_CTL) and with induction of HT (WT + HT), as well as EDNRA rats without (EDNRA_CTL) and with induction of HT (EDNRA + HT). A total of N = 59 rats were observed; N = 28 rats served as the control group (CTL group), and HT were induced in N = 31 rats (HT group) (Table 1).

The animals had access to food and water ad libitum. After randomization, the rats were held in a single cage with a 12-h light–dark cycle. The observation period was 12 weeks [17]. The animals that died during the surgery or observation period were excluded from further investigations.

The rats in the CTL group did not undergo sham surgery. Surgeons were blinded for the genotype of the rats during surgery, noninvasive blood pressure (NIBP), and brain preparation and microscopy. Investigators were blinded for immunofluorescence and functional investigations.

### 2.2. Genotyping

Rat DNA was isolated from ear punches by proteinase K (0.5 mg/mL) digestion in a TENS buffer (50 mM Tris pH 8.0, 100 mM EDTA pH 8.0, 100 mM NaCl, 1% SDS) overnight at 55 °C. After protein precipitation with a quarter volume of saturated NaCl and centrifugation, DNA was precipitated from a supernatant with one volume of isopropanol, washed once with 70% ethanol, and resuspended in 10 mM Tris pH 8.3.

DNA (50 ng) was used for PCR amplification in a 1× buffer with 0.25 µM primer each (EDNRA Lg del F: CCAGGGTTGGGAGATTTCTT and EDNRA Lg del R: GAGTGGGGAGGAGAGAAACC), 0.1 mM dNTPs, 1.5 mM MgCl_2_, 0.15 µL Platinum Taq polymerase, and the following protocol: 5 min at 94 °C, 30× (30 s at 94 °C, 30 s at 60 °C, and 2 min at 72 °C), and 7 min at 72 °C. Wild-type (1504 bp) and mutated (1092 bp) fragments were separated in 0.8% agarose gels (Figure 1).

### 2.3. Noninvasive Blood Pressure Measurement

The measurement of the blood pressure (BP) was performed by the tail-cuff method. This method is well-known and validated to resemble invasive measurements [29]. The animals were restrained with an opening for their tails. A blood pressure cuff with a distal pulse sensor was placed at the tail. The animals were exposed to an infrared light to induce heat exchange by the tail artery and enable the NIBP measurement. The systolic BP and pulse were measured with the NIBP-System for rats IN 125/R (ADInstruments Ltd., Oxford, UK) and recorded with LabChart 7 (ADInstruments Ltd.). This method was trained with all rats over a period of 2 weeks prior to the start of experiments to relieve iatrogenic stress. NIBP measurements were then performed, initially before surgery (week 0), around week 4, week 8, week 10, and before euthanasia at week 12.

### 2.4. Induction of Arterial Hypertension (HT)

All the animals undergoing the surgery were anesthetized with an intraperitoneal application of a mix of ketamine (Ketamin 10%, MEDISTAR Arzneimittelvertrieb GmbH, Ascheberg, Germany) in a dose of 100 mg/kg body weight and xylazine (Xylazin 2%, Ceva Tiergesundheit GmbH, Düsseldorf, Germany) in a dose of 10 mg/kg body weight. Body temperature was maintained at 37 °C using a heating pad. The rats were shaved at the neck and abdomen and disinfected with Braunol^®^ (B. Braun Melsungen AG, Melsungen, Germany).

The following surgery was performed with the aid of a Leica M80 microscope (Leica Mikrosysteme Vertrieb GmbH, Wetzlar, Germany). A median laparotomy was performed, and the intestines were put in a wet gauze compress to avoid evaporation. The dorsal ramus of the renal artery was identified on both sides and occluded with a non-absorbable suture (Premilene^®^ 10-0, Aesculap AG, Tuttlingen, Germany). The effectiveness of this procedure was directly observed by livid coloring of the superior kidney pole. Guts were put back in their anatomical position, and the intraperitoneal space was filled with sterile 0.9% saline (B. Braun Melsungen AG). The wound was sutured with two absorbable sutures (Safil^®^ 3-0 and Monosyn^®^ 4-0, Aesculap AG).

In the second step, the animals were kept in the supine position, and a median ventral incision of the neck was performed. Common carotid arteries (CCAs) were located on both sides and occluded with a non-absorbable suture (Premilene^®^ 10-0, Aesculap AG) to regulate the cranial blood flow only by the posterior circulation. The wound was sealed with an absorbable suture (Monosyn^®^ 4-0, Aesculap AG). In total, the surgery lasted 90 to 120 min.

Postoperatively, the animals underwent analgetic treatment for 5 days with 1.75 mg/mL metamizole (Novaminsulfon^®^, Ratiopharm GmbH, Ulm, Germany) in the drinking water ad libitum.

After day 6, a high-saline diet was started with a dose of 1% saline drinking water ad libitum to intensify the development of HT. The individual steps of this modified model were previously described [30,31].

### 2.5. Brain Preparation

After the last observation and NIBP measurement, the rats were euthanized by exsanguination under inhalation of isoflurane. The brain was freshly extracted and observed by two neurosurgeons under a Leica M80 microscope with the purpose of finding anomalies such as IAs and ectasias in the circle of Willis. Observation images were taken by an intercalated Leica IC80 HD camera (Leica Mikrosysteme Vertrieb GmbH).

### 2.6. Immunofluorescence

The brainstem with the basilar artery (BA) of WT_CTL (N = 9), EDNRA_CTL (N = 8), WT + HT (N = 3), and EDNRA + HT (N = 6) rats was isolated and stored in 4% paraformaldehyde. After fixation for 48 h, the BAs were imbedded in agar and cut into 50 µm thick tissue sections with a Leica VT1000S vibratome (Leica Mikrosysteme Vertrieb GmbH). The tissues were blocked with 3% bovine serum albumin for 45 min at room temperature (RT). The polyclonal sheep–anti-EDNRA antibody (Antibodies-Online Cat# ABIN285478, RRID:AB_10779812) was incubated in a dilution of 12.4 µg/mL with 0.6 µL/mL Triton X at 4 °C overnight. The next day, three washing steps with phosphate-buffered saline (PBS) were provided. Then secondary antibody Alexa Fluor^®^ 568 donkey anti-sheep (Abcam Cat# ab175712, RRID:AB_2892984) was incubated at a dilution of 10 µg/mL for 1 h at RT followed by another three washing steps with PBS. Nuclear counterstaining was performed with Hoechst 33342 (B2261, Sigma-Aldrich, Schnelldorf, Germany) in a dilution of 10 µg/mL for 15 min at RT followed by a last washing step with PBS. Sections were mounted in a VECTASHIELD^®^ Antifade Mounting Medium (Vector Laboratories, Burlingame, CA, USA).

Imaging was performed with a Leica SP8 AOTF confocal microscope (Leica Mikrosysteme Vertrieb GmbH) using a 40× water lens. One representative image per section was analyzed. The images were handled using LAS X (RRID:SCR_013673) and Fiji (RRID:SCR_002285). The mean fluorescence of the vessel wall, without the surrounding tissue and lumen, was analyzed by the mean gray value (MGV)—i.e., the sum of the gray values of all pixels in the ring segment divided by the number of pixels—of the bound EDNRA.

The negative control group of ring segments without the application of the primary antibody showed a significantly lower MGV in comparison with the ring segments of the same rats that received the primary antibody.

### 2.7. Functional Investigations

The excised brains along with the cerebral vessels of WT_CTL (N = 5), EDNRA_CTL (N = 6), WT + HT (N = 5), and EDNRA + HT (N = 5) rats were immersed in a cold modified Krebs–Högestätt solution containing the following components in mM: NaCl, 119; KCl, 3.0; NaH_2_PO_4_, 1.2; CaCl_2_, 1.5; MgCl_2_, 1.2; NaHCO_3_, 15; and glucose, 10. BA was dissected from the brainstem and cut into four ring segments, each approximately 2 mm in length. As already described elsewhere [25,26], ring segments were meticulously mounted on two L-shaped stainless-steel rods (70 mm diameter) in an organ bath (IOA-5301; FMI Föhr Medical Instruments GmbH, Seeheim, Germany) filled with the modified Krebs–Högestätt solution for the measurement of isometric force. The isometric force was determined by using a transducer (GM Scaime, Annemasse Cedex, France) and was digitally recorded (AMON FMI VitroDat 3.4 Software, FMI GmbH).

After the segments had been mounted, the temperature in the organ baths was gradually increased to 37 °C. During this adaptation period of 60 min, the segments were repeatedly stretched and adjusted to a resting tension of 3 to 4 mN. At the end of the adaptation period, a contraction was induced by a 124 mM K ^+^ Krebs solution (Krebs–Högestätt solution with an equimolar exchange of NaCl by KCl); this was used as the reference contraction for subsequent investigations. At the end of each experiment, the application of the 124 mM K ^+^ Krebs solution was repeated. Segments that reached less than 2.5 mN in the first contraction or developed less than 85% of the tension reached at the beginning were discarded from further analysis. The functional integrity of the endothelium was tested by an application of acetylcholine (10^−4^ M) (Sigma-Aldrich) after a precontraction had been induced with 5-hydroxytryptamine (10^−5^ M) (Sigma-Aldrich). A relaxation of more than 30% of the precontraction indicated a functionally intact endothelium.

Concentration–respond curves (CRCs) for the ET-1-induced (endothelin-1, Calbiochem–Novabiochem, Bad Soden, Germany) contraction were constructed in segments under resting tension. ET-1 was cumulatively applied (concentration range 10^−11^ M to 3 × 10^−7^ M). Contraction was measured in millinewton force and is given as the percentage (contractility) of the reference contraction. For each completed CRC, we determined the E_max_ that was obtained and performed a linear regression analysis after the logarithmic transformation of the concentrations above and below the EC_50_ to calculate the negative decadic logarithm of the EC_50_, the pD_2_, as the measurement of the potency of ET-1.

### 2.8. Statistical Analysis

Data were entered into a computerized database (IBM SPSS Statistics for Windows, Version 27.0. Armonk, NY, USA: IBM Corp). The BP values, MGVs, as well as contraction values are expressed as mean ± standard deviation (SD). The Shapiro–Wilk test was used to test for normality. The homogeneity of variances was asserted using Levene’s test. If the data were normally distributed and the homogeneity of variance was met, the differences between the four groups were evaluated with one-way analysis of variance (ANOVA) with Tukey post hoc analysis. If the data were not normally distributed or the assumption of the homogeneity of variances was violated, Welch-ANOVA with Games–Howell post hoc analysis was performed. Mauchly’s sphericity test was used to test the assumption of sphericity. If homoscedasticity was met, the longitudinal data of the BP of each group were analyzed by repeated-measures ANOVA. Statistical significance was assumed when *p* < 0.05. If the quantity was small (e.g., N < 5), no statistical tests were applied.

The vascular anomalies in the circle of Willis were classified as “no anomalies”, “ectasia” (including fusiform aneurysms and nascent aneurysms), and “saccular aneurysms”. A χ^2^ test was used with an assumed significance when *p* < 0.05.

## 3. Results

### 3.1. Mortality

Complete lethality in utero was observed for homozygous offspring from heterozygous breeder pairs with EDNRA knockout mutation, in three independent lines generated at SAGE with varying deletion sites and lengths upon zinc finger nuclease technology, so heterozygous mutants were assessed in this study as EDNRA rats. No rats out of the CTL groups died during the observation period. About 23% of WT + HT and 17% of EDNRA + HT died during the observation period. There was no significant difference between WT + HT and EDNRA + HT mortality. In autopsy, neither cerebral vascular pathology nor spontaneous SAH was detected. This yields an overall mortality of 19% for the induction of HT in this model.

### 3.2. Noninvasive Blood Pressure Measurement

The four groups showed a statistically significant difference for the different time points (week 4: *F*(3, 51) = 75.52, *p* < 0.001; week 8: Welch’s *F*(3, 25.65) = 62.80, *p* < 0.001; week 10: *F*(3, 49) = 43.69, *p* < 0.001; week 12: Welch’s *F*(3, 16.21) = 77.53, *p* < 0.001). The results of the Tukey post hoc analysis for weeks 4 and 10 as well as the Games–Howell post hoc analysis for weeks 8 and 12 revealed a significant difference between every group at every time point except between both CTL groups at all time points (Appendix A, Figure 2).

The repeated-measures ANOVA for every single group revealed no statistically significant difference for the different time points for WT_CTL, EDNRA_CTL, and EDNRA + HT. The repeated-measures ANOVA for WT + HT determined that the mean BP showed a statistically significant difference between measurement time points, *F*(3, 18) = 7.96, *p* = 0.001. However, the results of the post hoc analysis revealed no significant difference between those time points for WT + HT.

### 3.3. Morphological Changes in the Circle of Willis

No vascular anomalies or IAs were found in the CTL groups. Vascular anomalies as well as IAs (Figure 3) were observed only in WT + HT and EDNRA + HT but showed no significant difference between those two groups, χ^2^(1) = 1.33, *p* = 0.250, φ = 0.23 (Figure 4 and Table 2).

### 3.4. Immunofluorescence

Immunofluorescence was performed to visualize EDNRA biosynthesis. A total of 134 ring segments of BAs were investigated (Appendix A). The fluorescence of the vessel wall representing the EDNRA biosynthesis was embodied by the MGV. WT + HT was excluded from further analysis as the group consisted of N = 3 animals. There was no statistically significant difference in the MGV for the other three groups (WT_CTL, EDNRA_CTL, and EDNRA + HT), Welch’s *F*(2, 51.53) = 0.88, *p* = 0.420 (Figure 5).

### 3.5. Functional Investigation

After cumulatively applying ET-1 to the BA segments, a contraction could be observed in all four groups. Maximum contractility was reached at a concentration of 10^−7^ M ET-1. The contractility to ET-1 of BA segments showed a statistically significant difference for the four groups, *F*(3, 46) = 3.63, *p* = 0.020. The results of the Tukey post hoc analysis revealed a significant difference (*p* = 0.011) between contractility of WT + HT and EDNRA + HT (26.36, 95%-CI[4.79, 47.94]) (Appendix A, Figure 6). Without induction of HT, no difference was seen in contractility of both WT_CTL and EDNRA_CTL.

After completing the CRC for all groups, the pD_2_ as measurement of potency of ET-1 could be calculated for each group. The potency of BA segments showed a statistically significant difference for the four groups, Welch’s *F*(3, 20.91) = 4.13, *p* = 0.019. The results of the Games–Howell post hoc analysis revealed a significant difference (*p* = 0.030) between the potency of EDNRA_CTL and EDNRA + HT (0.26, 95%-CI[0.02, 0.51]) (Appendix A).

## 4. Discussion

Despite the high morbidity, mortality, and, therefore, socioeconomic importance of SAH, the underlying pathogenesis and pathophysiology of IAs are still not completely understood [33]. Many in vivo models were developed to investigate risk factors and treatment options. Mostly used are combined rat models with surgical preparation and additional drug intake (e.g., NaCl, BAPN, elastase, corticosterone, or angiotensin) [17]. The first and frequently modified model was established in 1978 by Hashimoto et al. via the ligation of unilateral common carotid artery, BAPN, and, later on, NaCl intake to induce human-like IAs [31,34]. The complex mechanisms of endothelial dysfunction provoked, among other things, by both HT and WSS are the recent and auspicious objects of research in IA pathogenesis [11].

The advantage of mouse models is the possibility of genetic variation, which is poorly used in rats. However, rat models are easier to handle especially for surgical interventions. The increasing role of genetics in IA pathogenesis could be corroborated by the identification of single nucleotide polymorphisms near the EDNRA gene that are associated with IAs [18,35]. Therefore, the aim of this study was to evaluate an experimental model of HT and IA induction that avoids many potentially interacting substances such as BAPN or elastase, thus investigating the role of the EDNRA in IA pathogenesis.

After creating a completely novel EDNRA-deficient rat line, the importance of EDNRA in the organism was put in evidence by the intrauterine death of homozygous rats. This finding is in good agreement with the observations in mice with homozygous EDNRA absence that died shortly after birth [36]. Thus, a homozygous rat EDNRA knockout line was not achievable, and only heterozygous rats could be used for this study. The deletion of 412 bp was verified by PCR genotyping of each rat.

Both WT + HT and EDNRA + HT showed an overall mortality of 19% through the induction of HT (surgery and saline diet). This observation matches with a previously described mortality of 17% in an analogous experiment [30]. Importantly, mortality did not significantly differ between WT and EDNRA rats. None of WT_CTL and EDNRA_CTL died during the observation period.

Measurements of the systolic BP confirmed the successful induction of HT in both EDNRA + HT and WT + HT. There were significantly higher BP values at all time points during the observation period in comparison between the CTL and HT groups. The HT groups had a steady HT after induction, but EDNRA + HT had even higher BP levels after induction. The BP of the CTL groups did not differ.

IAs or vascular anomalies were found neither in WT_CTL nor EDRNA_CTL. IAs could be induced in 3 out of 25 rats in the HT groups. There were more saccular IAs and vascular abnormalities that could evolve into IAs in EDNRA rats during the 12-week observation period, even though this effect was not significant. The overall frequency of IA induction did not differ from previous studies, also validating this model [37,38,39,40].

As EDNRA_CTL did not develop any vascular abnormalities as well as no notable HT, IA induction may not be primarily based on the EDNRA deletion but on induced HT. Regarding the IA development, the influence of HT seems to be more powerful than the genetic modification within the EDNRA gene alone. Nevertheless, this study proved that EDNRA rats could cope less when HT was induced in them. This may secondarily lead to an increased IA induction.

To examine the histological changes in the vessel wall provided by the WSS and HT, as well as the impact of the EDNRA, an immunofluorescence analysis was realized. For this purpose, an antibody against EDNRA was used with a counterstaining of nuclei by HOECHST. WT + HT were not statistically analyzed due to the sample size. All other groups (WT_CTL, EDNRA_CTL, and EDNRA + HT) showed equal values of fluorescence as a marker of conjugated EDNRA in the vessel wall. Thus, the 50% reduction of EDNRA biosynthesis may be partially compensated by slowed membrane turnover and degradation of EDNRA. Immunocytochemistry has notorious limits to detect fine downregulations, and also quantitative immunoblots are known to reliably detect only twofold effects, so technical issues probably explain why the genotype effects were not validated by this approach. There was a higher albeit not significant level of EDNRA fluorescence in the vessel wall in the HT groups compared with the CTL groups. Sauvageau et al. already reported similar findings that showed a higher but also not significant protein expression of the EDNRA after HT induction in WT rats [41].

If steady-state abundance of EDNRA appears not significantly reduced by immunodetection, but the phenotypical characteristics of EDRNA rats—shown by the different BP induced by the hemodynamic stress—are essentially different, the function of the EDNRA has to be affected by the genetic modification. For the investigation of the EDNRA function as the most powerful constrictor of human vessels, a functional investigation of rat BA segments was performed [22]. The most effective concentration for all four groups was seen at 10^−7^ M of ET-1. Both WT_CTL and EDNRA_CTL showed a similar maximum contractility. WT + HT showed an intensified contractility, whereas EDNRA + HT showed a reduced contractility. The increased contractility of the vessels of WT + HT reflects the usual reaction to HT [42,43]. The reduced contractility of the vessels of EDNRA + HT indicates a dysfunctional EDNRA due to the mutation. The potency of ET-1 was equal in WT_CTL and WT + HT, as expected. Similar with contractility, the potency of ET-1 was also affected by the mutation. EDNRA_CTL showed a higher potency of ET-1. As less ET-1 finds docking sites to activate EDNRA, the remaining ET-1 may activate endothelin receptor type B (EDNRB), which moderates vasodilatation and antagonizes the effects of EDNRA. This could explain the normotension of EDNRA_CTL rats that are not stressed. Due to the hemodynamic stress, the potency of ET-1 in EDNRA + HT reaches the level of that in WT rats. The underlying mechanisms remain unclear, and further studies are needed.

In summary, the abundance of EDNRA protein appears not detectably dysregulated by the genetic modification of heterozygous rats, but the functional disturbance of contractility and potency of ET-1 validates the biological impact of the targeted mutation and leads to the described phenotypical occurrence of dysregulated adaption to the hemodynamic stress. Nevertheless, the underlying mechanisms of dysregulation and IA development, growth, and rupture still remain unclear after this 12-week observation.

Our study has the following limitations: The sample size of the HT-induced rats was small. Secondly, a longer observation period, possibly across the entire lifespan of EDNRA rats, could provide more IA induction as many rats already developed vascular ectasia. Moreover, additional staining would provide more data on the interaction between EDNRA and EDNRB.

## 5. Conclusions

In this study, we could elucidate the important role of endothelin receptors in the pathogenesis of hypertension and IA formation. This completely novel in vivo model of hypertension and possibly IA induction can be the basis for further studies, which may include investigations of EDNRB as well as the observation of the outcome of SAH followed by IA rupture. A better understanding of the underlying pathomechanisms is necessary for novel therapy options for HT and SAH and may be possible by selecting an important subgroup at risk.

## Figures and Tables

**Figure 1 brainsci-12-01239-f001:**
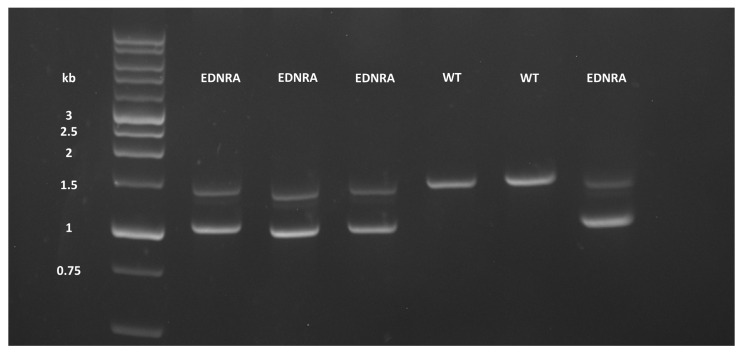
Exemplary results of genotyping of six rats’ DNA. Heterozygous EDNRA rats showed wild-type bands and mutant bands (1504 bp and 1092 bp) as verification of deletion. Marker on left side, kb = kilobase pairs.

**Figure 2 brainsci-12-01239-f002:**
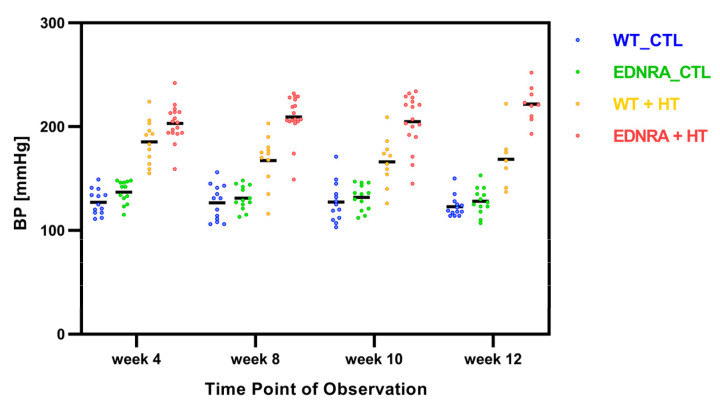
Comparison of individual data points and mean BP values (black bars) in mmHg of four groups at different time points of observation (see text and Appendix A for detailed data).

**Figure 3 brainsci-12-01239-f003:**
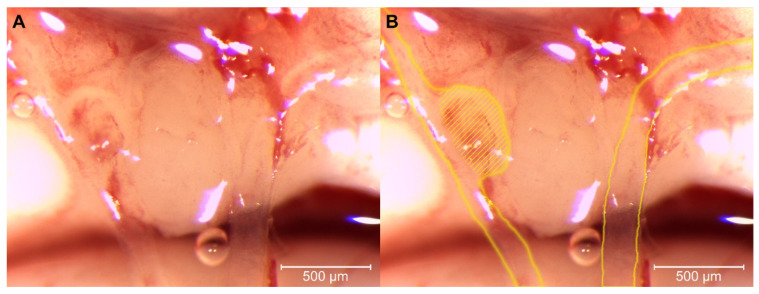
Light microscopy (magnification: 50×) of the brain of EDNRA + HT rat after observation period of 12 weeks. The basilar artery (not pictured) branches into both posterior cerebral arteries [32]. A saccular aneurysm with detritus evolved at the right posterior communicating artery. (**A**) Original picture. (**B**) Marked left and right posterior communicating artery with a saccular aneurysm (hatching).

**Figure 4 brainsci-12-01239-f004:**
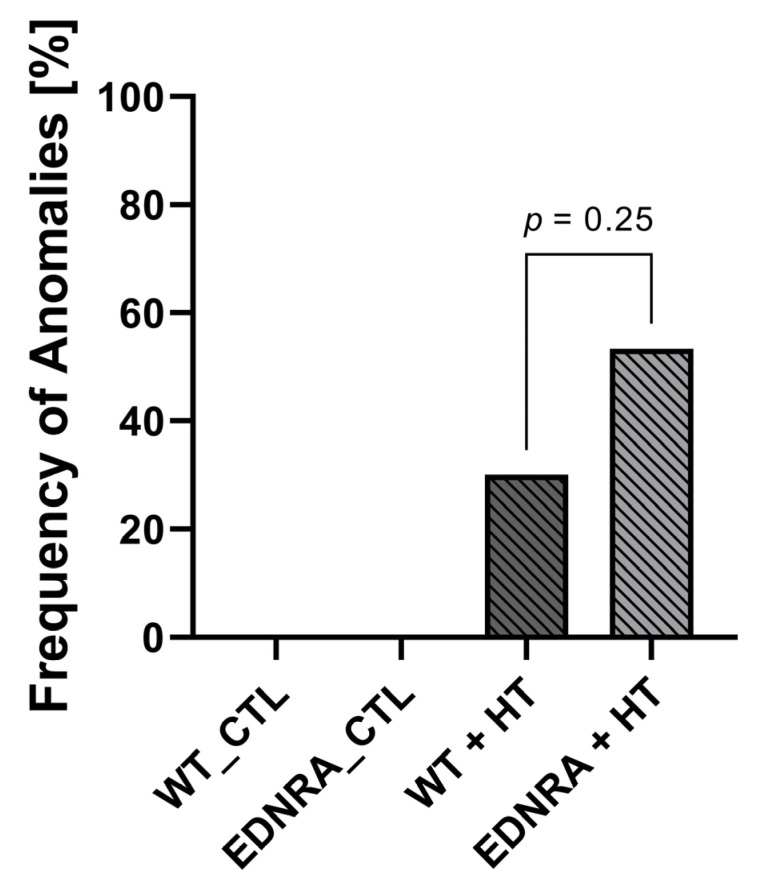
Vascular anomalies (ectasia and aneurysm) were observed only in HT-induced rats (*p* < 0.001) after observation period of 12 weeks. WT_CTL (N = 0/14) and EDNRA_CTL (N = 0/14) did not develop vascular anomalies. There was no difference of frequency of anomalies between WT + HT (N = 3/10) and EDNRA + HT (N = 8/15).

**Figure 5 brainsci-12-01239-f005:**
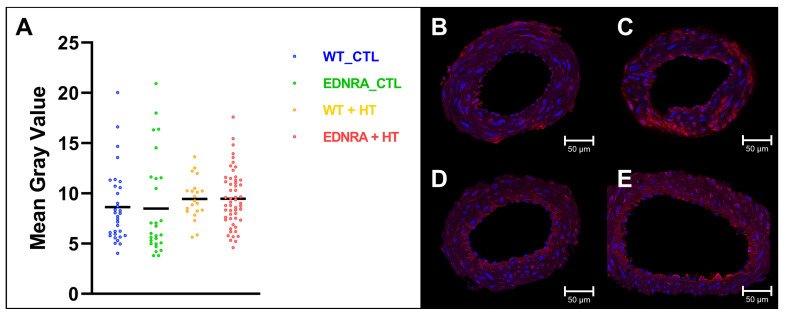
(**A**) Individual data points and means (black bars) of mean gray value of EDNRA in immunofluorescence after observation period of 12 weeks. Both CTL and HT groups showed similar levels of fluorescence intensity as reflection of EDRNA abundance. Despite tendency to higher levels after HT induction, none of the groups significantly differed. Therefore, EDNRA abundance seemed not to be higher after HT induction, neither in WT rats nor EDNRA rats. (**B**–**E**) One exemplary ring segment of BA of each group showing EDNRA immunoreactivity (red) and nuclei (blue): (**B**): WT_CTL; (**C**): EDNRA_CTL; (**D**): WT + HT; and (**E**): EDNRA + HT.

**Figure 6 brainsci-12-01239-f006:**
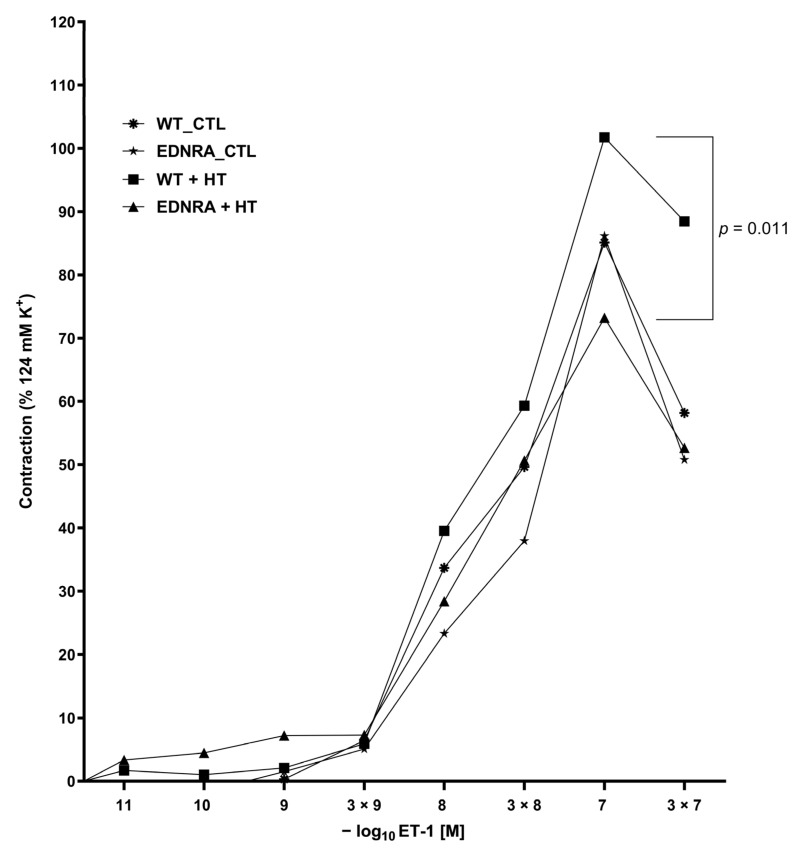
Concentration–response curves (CRCs) for ET-1-induced contraction of BA segments of each group (WT_CTL (N = 9), EDNRA_CTL (N = 19), WT + HT (N = 11), and EDNRA + HT (N = 11)) after observation period of 12 weeks. Maximum contractility was similar in CTL groups, increased in WT + HT (square), and decreased in EDNRA + HT (triangle). Maximum contractilities of WT + HT and EDNRA + HT were significantly different (*p* = 0.011). Potency of ET-1 was significantly increased in EDNRA_CTL, whereas potency of EDNRA + HT was equal to both groups of WT rats (see text and Appendix A for detailed data).

**Table 1 brainsci-12-01239-t001:** Distribution of rats in four experimental groups.

	Control Group (CTL)	Induction of HT
**WT**	14	13
**EDNRA**	14	18

WT rats without induction of HT (WT_CTL), EDNRA rats without induction of HT (EDNRA_CTL), WT rats with induction of HT (WT + HT), and EDNRA rats with induction of HT (EDNRA + HT).

**Table 2 brainsci-12-01239-t002:** Absolute and relative (in percentage) incidence of vascular anomalies.

	N	No Anomalies	Ectasia	Aneurysm
**WT_CTL**	14	14 (100%)	0 (0%)	0 (0%)
**EDNRA_CTL**	14	14 (100%)	0 (0%)	0 (0%)
**WT + HT**	10	7 (70%)	2 (20%)	1 (10%)
**EDNRA + HT**	15	7 (47%)	6 (40%)	2 (13%)
***p*-value**		0.25	0.29	0.8

N = number of rats that survived the observation period of 12 weeks and were able to admit to brain preparation.

## Data Availability

Data are available upon reasonable request from the authors.

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
