# Peer review of "Experimental Induction of Intracranial Aneurysms in Rats: A New Model Utilizing a Genetic Modification within the EDNRA Gene"

_brainsci, 2022, doi:10.3390/brainsci12091239_

Round 1

Reviewer 1 Report

The manuscript, which title is “Experimental induction of intracranial aneurysms in rats: A 2 new model utilizing a genetic modification within the EDNRA gene” is interesting and write well. There is one question in the manuscript. The authors should provide the Histopathological results for support their hypothesis in the present study.

Author Response

Thank you for your suggestions. We established a new in-vivo model without additional drug application that interrupts collagen cross-linking for example. Primary effect of the genetically modification seems to be the dysregulated hypertension. We therefore performed the functional analyses.

We also performed the immunofluorescence to visualize and determine the EDNRA biosynthesis. We revised Figure 5 by adding 3 more exemplary images (5C-5E) of ring segments to display every group (we adjusted brightness to improve visibility of the fluorescence). Therefore, we adjusted the figure legend as well.

The reviews encouraged us to do further studies to investigate the underlying pathophysiology. Therefore, based on this study we plan to approve another study at our local ethics committee to perform more specific and detailed investigations like immunofluorescence of other receptors (like EDNRB), histopathological staining, or western blot.

We believe that after incorporating the issues listed above, the manuscript is clearer and the study is now better understood. We hope that the manuscript now is eligible for publication.

Reviewer 2 Report

This article about a A new rat model utilizing a genetic modification within the EDNRA gene is interesting, well written but remains really superficial leading to a misunderstanding of what this article is.

Minor:

Introduction:

this section could be more developed to give the context and why a new model is needed

Please add the hypothesis of the study

Fig3 A-B: could you add scale bar and magnification in the legend

Fig 4, could you add the n for each group

Major:

"3.4 Immunofluorescence" this sections has to be re-written. 

The purpose of it is not clear, I don't really understand what am I looking at, what the staining is, where are we located,....

3.5 section, I would suggest to introduce the results and add context. In the graph is not really clear who is statistically different from others. 

Since it's a new model, I would highly recommend to further characterize the model, using blood flow measurement, are the blood vessels altered?, how about the blood brain barrier? ...

I would suggest to dig deeper to really characterize the model and make a clear link between the receptor and the disease. At this stage the MS is too superficial, and the presentation of figures can be improve. 

Author Response

Thank you for the useful comments. We believe that after incorporating the issues listed below, the manuscript is clearer and the study is now better understood. We hope that the manuscript now is eligible for publication.

We revised the introduction to clarified the necessity of this model and our hypothesis (line 57 ff): “Most of the used in-vivo models utilize a combination of surgical preparation and additional drug intake (e.g. NaCl, β-aminopropionitrile (BAPN), elastase, corticosterone or angiotensin) [17]. Especially BAPN and elastase weaken the vessel wall, not allowing further histological investigations of a natural IA development. We hypothesize that combination of HT and modification within the EDNRA gene aggravates the HT in such a way IA development is induced – making the intake of interfering drugs superfluous. This may offer an opportunity for further investigations in a natural setting without alteration of the vessel wall.

We revised Figure 3 by adding a scale bar and stated the magnification in the legend of Figure 3: “Light microscopy (magnification: 50x) of the brain of an EDNRA+HT rat after the observation period of 12 weeks. A saccular aneurysm evolved at right posterior communicating artery. The BA (not pictured) branches into both posterior cerebral arteries.[32] A saccular aneurysm with detritus developed at the right posterior communicating artery”

We stated the N for each group in the legend of Figure 4: “Vascular anomalies (ectasia and aneurysm) were observed only in HT-induced rats (p < 0.001) after observation period of 12 weeks. WT_CTL (N = 0/14) and EDNRA_CTL (N = 0/14) did not develop vascular anomalies. There was no difference of frequency of anomalies between WT+HT (N = 3/10) and EDNRA+HT (N = 8/15).”

We revised the section “3.4. Immunofluorescence” to clarify the intention and given data of the immunofluorescence (line 288 ff): “Immunofluorescence was performed to visualize EDNRA biosynthesis. 134 ring segments of BAs were investigated (Table S2). The fluorescence of the vessel wall representing the EDNRA biosynthesis was embodied by the MGV. WT+HT was excluded from further analysis as the group consisted of N = 3 animals. There was no statistically significant difference in MGV for the other three groups (WT_CTL, EDNRA_CTL and EDNRA+HT), Welch’s F(2, 51.53) = 0.88, p = .420. (Figure 5).” Moreover, we revised Figure 5 to clarify that the colored dots in Figure 5A represent the different groups based on Figure 2; we added 3 more exemplary images (5C-5E) of ring segments to display every group (we adjusted brightness to improve visibility of the fluorescence). Therefore, we adjusted the figure legend as well.

We revised Figure 6 by adding a bar to show the significant difference in contractility between WT+HT and EDNRA+HT. The figure legend was adapted, too. Moreover, we revised the section “3.5. Functional investigation” to introduce the results of the functional investigation (line 303 ff): “After applying ET-1 to the BA segments cumulatively, a contraction could be observed in all four groups. Maximum contractility was reached at a concentration of 10-7 M ET-1. Contractility to ET-1 of BA segments differed statistically significant for the four groups, F(3, 46) = 3.63, p = .020. Tukey post-hoc analysis revealed a significant difference (p = .011) between contractility of WT+HT and EDNRA+HT (26.36, 95%-CI[4.79, 47.94]) (Table S3, Figure 6). Without induction of HT, no difference was seen in contractility of both WT_CTL and EDNRA_CTL. After completing the CRC for all groups, the pD2 as measurement of potency of ET-1 could be calculated for each group. Potency of BA segments differed statistically significant for the four groups, Welch’s F(3, 20.91) = 4.13, p = .019. Games-Howell post-hoc analysis revealed a significant difference (p = .030) between potency of EDNRA_CTL and EDNRA+HT (0.26, 95%-CI[0.02, 0.51]) (Table S4).”

The reviewer is absolutely right. Further investigations are of high interest and especially potential treatment options (e.g. antihypertensive drugs) are the next steps that we are looking for. For the first description of the model, all available specimen were used for either functional investigation or immunofluorescence. As arteries would need to be processed freshly for most of these analyses, we are not able to do so in accordance with animal protection laws.

Primarily, aim of the current study was to establish and validate this new in-vivo model. We included first investigations on the influence of this genetic modification within EDNRA gene on IA pathogenesis. The reviews encouraged us to do further studies to investigate the underlying pathophysiology. Therefore, based on this study we plan to approve another study at our local ethics committee to perform more specific and detailed investigations like immunofluorescence of other receptors (like EDNRB), histopathological staining, or western blot.

Round 2

Reviewer 1 Report

No more question.

Reviewer 2 Report

Thank you for the modifications on the MS. 

It still remains superficial which is really a shame, since the subject is really interesting. 

Also, before the publication, i would recommend to modify the discussion section. At this point this section is just a wrap up of results.